# Relationships between Diabetes and the Intestinal Microbial Population

**DOI:** 10.3390/ijms24010566

**Published:** 2022-12-29

**Authors:** Stephen C. Bondy

**Affiliations:** 1Department of Medicine, Center for Occupational and Environmental Health, University of California, Irvine, CA 92697, USA; scbondy@uci.edu; 2Department of Environmental & Occupational Health, University of California, Irvine, CA 92697, USA

**Keywords:** microbiome, diabetes, intestinal bacteria, butyrate, diet

## Abstract

Diabetes is a metabolic disorder characterized by lower responsiveness of tissues to insulin and consequent large variations in circulating levels of glucose. This fluctuation has harmful effects as both hyperglycemia and hypoglycemia can be very injurious. The causes of diabetes are varied but the consequences are rather uniform. Dietary factors are important especially in adult onset type 2 diabetes (T2D) while type 1 diabetes (T1D) is characterized by having a stronger heritable component and involving autoimmune attach on pancreatic beta cells. This review is focused on the relation of the bacterial components found within the intestine, to the establishment and maintenance of diabetes. The precise composition of the gut microbiome is increasingly recognized as a factor in organismic health and its interaction with a variety of disease states has been described. This is especially marked in the case of diabetes since the nature of the diet is an important factor in establishing both the microbiome and the incidence of diabetes. The bidirectional nature of this relationship is discussed. The effects of disease that lead to altered microbiomal composition together with aberrant metabolic changes are also included. Emphasis is given to the important role of short chain fatty acids (SCFAs) as mediators of the microbiome-diabetes relation.

## 1. Introduction

The incidence of Type 2 diabetes (T2D) is growing rapidly worldwide [1]. To a large degree, this reflects the transition from consumption of less refined native foodstuffs toward a modern industrially produced diet rich in rapidly assimilable processed materials. This shift has also resulted in widespread gains in body weight, so that while in the past, obesity was associated with affluence, it is now more prevalent among lower income groups. While Type 1 diabetes (T1D) is generally found among a much younger age group, it is increasing equally rapidly as T2D. An over 100-fold variation of the extent of its incidence exists between various countries, being highest in Scandinavian countries and lowest in least developed countries [2]. This suggests that T1D prevalence is also strongly affected by environmental factors.

While the various forms of diabetes have a significant overlap in some aspects, the distinctions between them are briefly discussed in the next section.

The components of the diet, are a key determinant of a predisposition to diabetes. Diet is also an important factor in establishing the nature of the bacterial, viral and fungal elements that constitute the intestinal microbiome. The elements within the heterogeneous collection of microbiota are variable and in constant flux. Inappropriate composition of the gut microbiota and resulting abnormal content of soluble metabolic factors have been related to increases in adiposity, chronic inflammation and insulin resistance and imbalance of microbiome composition is common in pathological states such as obesity and metabolic syndrome, and diabetes [3].

The gut microbiomal composition both influences and is sensitive to, the state of health of the host. This review is focused on how the microbiome can influence the development and progression of diabetes, and conversely, how the disease may have an impact on the microbiome. The molecular processes that may underlie this reciprocal relationship are considered.

## 2. Type 1 and 2 Diabetes, Similarities and Distinctions

The two major types of diabetes have been classically distinguished in several general ways. Type 1 diabetes (T1D) is described as a disease with a large genetic component that involves an auto-immune attack on insulin producing pancreatic beta cells. It therefore tends to develop early in life and involves failure of insulin synthesis. T1D involves irreversible destruction of insulin producing cells, and autoantibodies to pancreatic b-cells can to detected long before the onset of clinical disease [4]. This disruption leads to wide fluctuations of levels of circulating glucose. Both hyperglycemia and hypoglycemia can rapidly lead to dangerous toxic effects. Disease onset can be very sudden. As T1D is essentially a deficiency disease, the most effective treatment is replacement therapy by administration of insulin and regular monitoring of blood glucose levels.

Type 2 diabetes (T2D) develops more slowly and progressively, generally later in life and may reflect lifestyle patterns of diet and exercise. In this disease insulin may still be produced but tissues become insulin resistant and do not respond to this hormone by adequately increasing uptake of circulating glucose. T2D is associated with excess body weight, physical inactivity, but additionally, has a genetic component. Epigenetic changes that can be regulated by metabolites produced by intestinal bacteria are likely important to establishing the onset of diabetes. The epigenetic profile determines the spectrum of proteins produced and this can influence the promotion or retardation of onset of the disorder. For example, histone deacetylases can be inhibited by butyrate produced by bacterial fermentation and can increase the degree of transcription of several genes associated with production of anti-inflammatory factors [5]. Likewise, expression of the microRNAs, miR-181a and miR-181b, which are associated with development of obesity and insulin resistance, is down-regulated by bacterially produced beneficial tryptophan metabolites [6].

In recent years, the degree of overlap between T1D and T2D has been increasingly recognized. Both have an association with the gene expression profile superimposed on which are environmental factors. Even among identical twins the extent of concordance of T1D is only 16% [7], while in the case of T2D it is 6% [8]. A degree of autoimmune attack on pancreatic insulin producing b-cells which is a key feature of T1D, is also found in T2D [9]. The age of onset of T1D is typically in childhood but can occur in adult life while T2D is also occasionally found in children. Other commonalities include the increased risk of either T1D or T2D following preterm birth [10] and genetic and molecular susceptibilities shared between the two variants [11]. Both variants of the disorder are also characterized by increased intestinal permeability and diminished bacterial diversity in the gut (discussed below).

This emerging convergence of perspectives on these variants of diabetes has led to identification of a new disorder reflecting a combination of their characteristics, latent autoimmune diabetes of adults (LADA) [12]. The identification of such an intermediate form has led to perception of diabetes as somewhat of a graded disease continuum.

## 3. Composition of the Human Microbiome and Relevance to Diabetes

The human intestinal tract has a surface area of around 300 m^3^, presenting a large interface with exogenous factors originating in the environment. A complex symbiotic relationship has evolved between these microbiota of which there are are over 2000 individual species, and the host. These vary considerably in composition during stages of maturation and also between geographic regions [13]. The breadth and variation of the intestinal microbiome is influence by many environmental factors. While the 12 phyla known to exist with the microbiome can be subdivided into many genera and a host of species, the focus of this list is upon those predominant phyla, members of which often bear an underlying similarity of their characteristics. Some distinctive and important species are also reviewed.

### 3.1. Firmicutes

The most predominant phylum in the adult human intestine are the Firmicutes. This phylum includes *Clostridium, Lactobacillus* and *Fecalibacterium* species, many of which are associated with production of butyrate and other short chain fatty acids (SCFA) which have been reported to attenuate inflammatory and allergenic activity [14].

#### Relevance to Diabetes

The content of differing Lactobacillus species in the microbiome in T2D can be differ dramatically. While the content of *L. acidophilus, L. gasseri, L. salivarius* are all increased in *T2D, L. amylovorus* is decreased [15]. Generalizations involving even a single genus cannot always be made but overall levels of butyrate -producing Firmicutes species are broadly decreased in T2D [16].

### 3.2. Bacteriodetes

The second most dominant phylum are Bacteriodetes which include several anaerobic *Bacteriodes* species such as *B. dorei, B. fragilis* and *B. thetaiotaomicron. Bacteroides fragilis* is a common cause of intra-abdominal infections in humans.

*Prevotella* is another genus found within this phylum which tends to promote inflammatory changes. T1D is associated with an elevated presence of Bacteriodetes

Non-vegetararians have a large excess of Bacteriodetes over Firmicutes species but this is ratio reversed in vegetarians [17].

#### Relevance to Diabetes

While butyrate is principally produced by Firmicutes, propionate and acetate are largely produced by Bacteroidetes [18]. Butyrate seems to be the most potent SCFA in attenuating inflammatory responses by reducing the accumulation of neutrophils and pro-inflammatory macrophages in the M1 conformation [19].

Levels of *B.dorei* have been used as s predictor for T1D onset [20].

Bacteriodetes and Actinobacteria are overrepresented in the microbiome of T2D [15] (Gurung et al., 2020). *B. thetaiotaomicron* reduced plasma glutamate concentrations in mice, and lowered diet-induced excess weight gain in mice [21].

### 3.3. Actinobacteria

The most abundant genus within actinobacteria is the *Bifidobacteria.* This genus constitutes around 5% of the adult microbiome but is over 90% of that of the infants under 2 years old. Species such as *Bifidobacterium adolescentis* can promote digestion of milk constituents and synthesize several B vitamins. Its administration has found therapeutic utility since 1899. *Bifidobacteria* are able to digest fiber to produce butyrate and also seem to suppress autoimmune activity. Lactobacillus and various Firmicutes (*Roseburia* and *Eubacterium*) also promote butyrate production [22].

#### Relevance to Diabetes

Inflammation is an important etiological factor for insulin resistance, which may lead to the development of T2DM [23] and several strains of Bifidobacteria have been found to inhibit inflammation by blocking NF-κB activation [24].

Fecal *L. acidophilus* content is elevated in uncontrolled diabetic patients [25]. Probiotic supplements that contain *Bifidobacterium* together with *Lactobacillus* strains may improve insulin sensitivity and reduces fasting levels of plasma glucose in diabetes [26].

### 3.4. Verrucomicrobiota

The phylum in represented in the gut by a sole species, *Akkermansia muciniphila* which colonizes the mucosal layer of the gut and is able to degrade mucin to acetate. This species is relatively prevalent in that it constitutes 0.5–5% of the total intestinal bacterial population.

#### Relevance to Diabetes

Obesity leads to a range of disease consequences among which diabetes is a major hazard. There is a clear association between obesity and T2D. Increased intestinal levels of *Akkermansia muciniphila* lead to inhibition of diet induced obesity and diabetes in mice [27]. Conversely, decreased abundance of this bacterium impairs glucose homeostasis [28]. The attenuation of diabetes in the presence of this this bacterium has recently been reviewed [27].

### 3.5. Proteobacteria

This phylum which constitutes less than 1% of the healthy human microbiome, includes the family Enterobacteriacea. An increased abundance of members belonging to this family, especially *Escherichia coli* a pathogen that has been associated with several inflammatory disease states including metabolic disorders and inflammatory bowel disease [29].

#### Relevance to Diabetes

An excess intestinal presence of *E. coli* at the expense of anti-inflammatory butyrate producing bacteria in T2D and in pre-diabetics has been reported on several occasions [30]. It has also been found in excessive amounts in T1D [31].

### 3.6. Bacillota

This phylum of anaerobes includes the *Blauta* genus, members of which have inflammation-suppressing qualities and antibiotic activity against specific pathogens [32] and *Ruminococcus* is important for digestion of resistant starches [33].

#### Relevance to Diabetes

Levels of these bacterial species within this phylum are elevated in T2D [15].

The content of butyrate-producing bacteria *Lactobacillus* (from the Bacillota phylum) is reduced in patients with T2D. These bacteria have also been reported to have anti-allergic effects [34] and to modulate gene expression in a desirable direction tending toward the alleviation of diabetes [35].

## 4. Relative Prevalence of Differing Bacterial Classes

An enrichment of pro-inflammatory bacteria (e.g., *Escherichia coli*) at the expense of anti-inflammatory bacteria (e.g., *Fecalibacterium prausnitzii)* in T2D has been reported on several occasions [30]. While it is clear that the predominance of various microbial species is altered in T2D, the causal relationships are unclear and whether these changes lead to T2D or are a response to the disorder is not well established. Furthermore, if the second possibility is the case, it is not firmly established if changes in microbiome composition represent a positive defensive response to the disorder or have a more damaging consequence [36]). Causality requires that when all confounders are allowed for, inducing change in one parameter should lead to modifications of a different parameter in a reproducible manner. These can only be shown by using delineated interventional experiments.

Much use has been made of the degree of prevalence of members of the Firmicutes phylum relative to that of those within Bacteriodetes. This widely used ratio (F/B) is reported as low in diabetes but high in obesity [37]. An increase in the F/B ratio has also been seen in obese mice [14]. Obesity and diabetes seem to be linked with different groups of intestinal microbiota [38]. In lean patients with T2D the F/B ratio is higher than in those with the more common form of the disease associated with excess fat deposition [39].

Since the severity of T2D in obese subjects is generally improved after weight loss, T2D in lean people may be a different disease from the diabetes associated with obesity. Malnutrition-related diabetes has several features that distinguish it from obesity-related diabetes [40]. Malnutrition is known to limit diversity of the gut microbiome but any association of this with diabetes has not been studied. The major defect in T2D of lean patients may involve a reduced capacity for insulin secretion rather than increasing insulin resistance [41,42].

Additionally, the number of contradictory reports concerning this use of the F/B ratio with respect to diabetes may be attributable to the existence of many potentially confounding factors such as selection of subjects, extent of physical activity, presence or absence of contemporaneous obesity [15]. In addition, disparities may exist due to methodological variations used. Taxonomic study of gut bacteria is often carried out performed by sequencing of 16S rRNA to allow precise characterization of microbial species. There can be variation of primers used in amplification of the 16S rRNA region, the sequencing technique, and analysis of data [43].

## 5. Bacterial Diversity

The appearance of diabetes-associated autoantibodies in T1D is preceded by an overall reduced diversity of the spectrum of intestinal microbiota [44]. Especially prominent is an elevated proportion of Bacteroides species [45,46] and a profound depression of *Akkermansia* [44]. The reduced overall palette of bacterial diversity combined with the disproportionate presence of Bacteriodetes may lead to abnormal development of the immune system and the appearance of immune related abnormalities including a trend toward increased autoimmune and inflammatory activity [47]. A similar situation whereby imbalance of specific bacterial species within the microbiome can promote autoimmune and inflammatory responses may exist in T2D [48]. Factors released from certain bacterial classes within the microbiome can be recognized and responded to by epithelial cells and immune cells, leading to a defensive inflammatory response by effector T lymphocytes. The extent of this reaction is delimited by regulatory T lymphocytes, induced by butyrate produced by several bacterial species especially those within the Firmicutes phylum. A distortion of the normal composition of the microbiome resulting from diminution of such Firmicutes, can lead to excess inflammatory activity. This information may reach the hypothalamus by way of the enteric nervous system and thereby lead to systemic hyperglycemia and insulin-resistance [49].

## 6. Bacterial Regulation of Intestinal Permeability

An increased intestinal permeability due to diminished integrity of tight junctions, enables dietary antigens and immune stimulants such as microbial lipopolysaccharides, to become systemically distributed. This can promote general inflammatory activity and specific autoimmune attack on the pancreatic β-cells. These damaging events are found in both T1D and T2D and are inhibited by butyrate [44,50]. This is enabled by stimulation of increased expression of a range of tight junction proteins. Inhibition of the activation of NLRP3 inflammasome and autophagy induced by LPS are inhibited by all SCFAs [51]. However, the potentially protective role of propionate and acetate remains uncertain since they have also been reported to have an opposite effect by impairing the assembly of tight junction proteins [44].

Increasing evidence suggests that gut leakiness is a cause rather than a consequence of the diabetic condition. Such leakiness has been reported to occur prior to the onset of T1D [52]. The hyperglycemia that often precedes overt T2D has been found to disrupt the tight adherence and barrier function of the intestinal epithelial cells [53].

## 7. Modulation of the Microbiome as a Means of Treating Diabetes

This strategy may be of especial value in T1D where pancreatic b-cells are lost before the onset of clinical symptoms of the disease [54]. Since T1D has a major genetic component, early detection of pre-diabetes can give a long period for initiation of attempted mitigation by modulation of the microbiome profile. T2D has also been improved by alteration in the intestinal microbiota [55]. Several means of enabling such changes are described below.

### 7.1. Diet

The constitution of the microbiome can rapidly by altered as a result of changes in the composition of the diet [56].

Dietary inclusion of modified starches can alter the numbers and activities of both autoreactive T cells and in this manner, may provide protection against autoimmune diabetes [45]. A causal relation between these events is likely but remains to be unambiguously demonstrated.

This diet rich in fibre and some plants compounds was associated with higher levels of *Akkermansia muciniphila* and *Faecalibacterium prausnitzii*, providing a reduction in endotoxemia in patients with T2DM [57]. The presence of non-digestible fibers, and complexes of starch and amylose in the diet are needed to serve as substrates to allow fermentation to butyrate. The useful qualities of butyrate are many fold and have been thought to include its acting as a histone deacetylase (HDAC) inhibitor or signaling through several G protein–coupled receptors (GPCRs) and consequent inhibition of NF-kB [58]. However, at lower concentrations, SCFA have recently been reported to activate a specific HDAC, acetyltransferase p300 [59]. Elevated levels of butyrate are associated with inhibition of NF-kB-associated histone deacetylases, leading to reduced activity of this inflammatory signaling pathway [60]. Short chain fatty acids including butyric acid (SCFA) are also reported to improve the function of the intestinal barrier which is impaired in both T1D and T2D. It may be that this is the major positive attribute of bacteria whose fermentation products include SCFA [61,62].

### 7.2. Exercise

Extended and regular exercise is known to improve glycemic control and to reduce systemic inflammation [63]. This is accompanied by reduced intestinal fungal mycetes overgrowth and intestinal permeability [64]. While exercise can increase microbiomal diversity [65], overall diet seems to be more important than exercise in establishing the composition and diversity of gut microbiota [66].

### 7.3. Pharmacological Agents

Metformin is an orally administered drug that is commonly used in T2D treatment. This agents acts by inhibition of hepatic gluconeogenesis. However, the molecular events underlying this are unclear [67]. Metformin reaches 300-fold higher concentrations in the gut than in the serum. Metformin treatment appears able to increase the proportion of *Akkermansia muciniphila* as well as butyrate-producing Bifidobacterium species [68], while decreasing the abundance of Bacteriodetes [56]. Its metabolic effects are likely to be mediated in part its actions in the intestine. It has been suggested that the predominant effect of metformin in regulating glucose levels, may involve decreasing the levels of *B. fragilis* and thus increasing content of the bile acid glycoursodeoxycholic acid leading to inhibition farnesoid X receptor (FXR) signaling [69]. T2D is characterized by microbiomal changes away from butyrate-producing taxa. The beneficial effects of metformin on T2D may be largely by increasing the relative abundance of microbes producing short-chain fatty acids [70].

The use of metformin has been sufficiently promising in studies on animals, that human studies are in progress [71].

A range of other pharmacological agents is emerging that involve indirect regulation of glucose levels. However, a significant portion of their utility is also likely to be based on their modification of the microbiome in an overall desirable direction [72]. These include the sulfonylureas which enhance insulin secretion by blocking ATP-sensitive K-channels, thiazolidinediones which increase sensitivity to insulin, alpha-glucosidase Inhibitors which retard the rate of oligosaccharide breakdown, and glucagon-like peptide-1 (GLP-1) receptor agonists which slows stomach emptying and increase insulin secretion. All of these also seem to drive the microbiome composition of the gut in a beneficial direction [72].

### 7.4. Probiotics

Many changes in the inflammatory profile, insulin resistance and bacterial composition in the gut of experimental animals have been reported following oral introduction of beneficial microorganisms such as Lactobacillus, Bifidobacterium or Akkermansia sp. into the diet. However, results from clinical studies have been more equivocal [73]. This is reminiscent of Alzheimer’s disease (AD) where rodent models of the disorder are easier to treat than are AD patients. The incomplete transference of success in animal models to humans may reflect the multifactorial nature of clinical disease and the much longer interval before manifestation of symptoms. These features probably result in the emergence of a more complicated disease profile in humans.

### 7.5. Fecal Transplantation

The introduction of bacterial species from healthy human to diabetic or pre-diabetic patients by this route attempts to improve the composition of the microbiome biome so as to create a profile with a more normal spectrum. Promising results have been obtained in experimental animals [74] and in human gestational diabetes [26]. Overall, there is growing evidence that the establishment of key bacterial species in the gut microbiome can lead an increased presence of SCFA as a result of fermentative activity [55]. The use of a single bacterial species (*Akkermansia muciniphila*) or even a specific membrane protein thereof, has been suggested as a more focused means of utilizing the desirable qualities of gut bacteria [75].

## 8. Summary

While this review has focused on beneficial products, notably SCFA, produced by the gut microbiome, it should be kept in mind that various classes of intestinal bacteria produce a range of harmful metabolites from dietary constituents and these are associated with obesity and diabetes. These include trimethylamine N-oxide produced from the choline and carnitine content of red meat and eggs, metabolites from the kyenurenine pathway of tryptophan breakdown, and imidazole propionate derived from histidine catabolism [76].

The close relation and many interaction between the diabetic state and the bacterial composition of contents of the intestine (summarized in Figure 1), offers a range of means of amelioration of this disorder. The incidence of diabetes can vary between different nations from as high as 30% of the population down to 5%. In addition, both childhood ands adult onset T1D also have a sharp variance between global regions [2]. This is somewhat surprising as genetic makeup plays a large role inT1D. It must be surmised that environmental factors are critical in determining the development of much T1D. Cultural and social determinants form the basis of most T2D. Thus, an inexpensive yet challenging means to dramatically alter the extent of the incidence of both types of diabetes can be by addressing and modifying extrinsic factors. The disease generally progresses slowly, giving much time for early interventions not involving sophisticated clinical medicine. The power of the intestinal microbiota in determining the state of organismic health is well illustrated in a recent report on kwashiorkor. Using animal models of this condition, a probiotic formulation served to optimize overall health despite severe malnutrition, as judged by reduction of weight loss, protection of immune function, improved resistance to infection and behavioral decline [77]. Addressing diabetes by way of revision of the microbiome spectrum is relatively non-invasive, and inexpensive. Furthermore, it does not involve extended drug usage which can at length induce further disruption of glucose homeostasis by provoking systemic responses contrary to those desired. However, improvements are only likely to be sustained by the patient’s development of a fresh lifestyle.

## Figures and Tables

**Figure 1 ijms-24-00566-f001:**
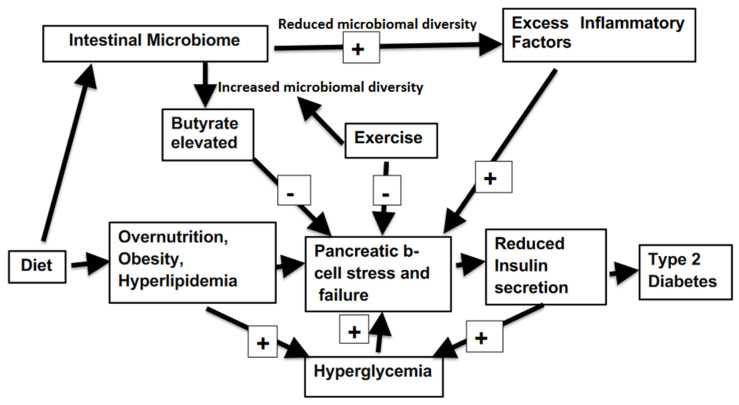
Factors influencing microbiomal interaction with diabetes progression.

## Data Availability

Not applicable.

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
