# Peer review of "Relationships between Diabetes and the Intestinal Microbial Population"

_ijms, 2022, doi:10.3390/ijms24010566_

Round 1

Reviewer 1 Report

This is an interesting and thought-provoking review that brings together information and research on intestinal microbial populations and their alterations in diabetes mellitus and its subtypes. It provides insights on the different means that may be used to either prevent and/or ameliorate the disorder  and also highlights the importance of diet and lifestyle on the intestinal microbiome and diabetes mellitus development.

The review is expansive and informative, but there were a few typographical/grammatical issues that need attention. Some of these are listed below:

Abstract, line 6:  change 'attach' to attack.

Page 1, introduction, paragraph 1, line 4:  change 'boy' to body.

Page 1, introduction, paragraph 1, line 8: insert the word 'of' after 'extent.

Page 1, introduction, paragraph 3, line 5:  delete the word 'composition'.

Page 2, under heading (2. Type 2...), paragraph 3, line 1:  delete the word 'those' before T1D and change the word 'as' to has.

Page 2, under heading (2. Type 2...), paragraph 3, line 6:  change 'age on onset' to 'age of onset'.

Page 2, under heading (2. Type 2...), paragraph 3, line 7: insert the word 'life' after 'adult.

Page 2, under heading (3. Composition...), paragraph 1, line 3:  insert the word 'there after 'which'.

Page 3, line 9:  the words 'are all increased in T2D' do not need to be italicised.

Page 3, line 33:  delete the the word 'This' before 'can' and continue the sentence.

Page 4, paragraph 1: the final sentence in incomplete.

Page 4, section 3.6, paragraph 2:  the reference (Gurung) is incomplete.

Page 4, section 4, paragraph 1, line 10:  change 'reproduce able ' to 'reproducible'.

Page 6, section 7.1, paragraph 3, line 6:  delete repeated words 'acting as'.

Page 6, section 7.1, paragraph 3, line 8:  The reference '(Lui)' is incomplete.

Page 6, section 7.3, paragraph 1, line 7: the reference '(Wu et al.,)' is incomplete. 

Author Response

The review is expansive and informative, but there were a few typographical/grammatical issues that need attention. Some of these are listed below:

Abstract, line 6:  change 'attach' to attack. Done

Page 1, introduction, paragraph 1, line 4:  change 'boy' to body.Done

Page 1, introduction, paragraph 1, line 8: insert the word 'of' after 'extent. Done

Page 1, introduction, paragraph 3, line 5:  delete the word 'composition'. Done

Page 2, under heading (2. Type 2...), paragraph 3, line 1:  delete the word 'those' before T1D and change the word 'as' to has. Done

Page 2, under heading (2. Type 2...), paragraph 3, line 6:  change 'age on onset' to 'age of onset'. Done

Page 2, under heading (2. Type 2...), paragraph 3, line 7: insert the word 'life' after 'adult. Done

Page 2, under heading (3. Composition...), paragraph 1, line 3:  insert the word 'there after 'which'. = Changed to "There are.."

Page 3, line 9:  the words 'are all increased in T2D' do not need to be italicised. Done

Page 3, line 33:  delete the the word 'This' before 'can' and continue the sentence. Done

Page 4, paragraph 1: the final sentence in incomplete. Corrected

Page 4, section 3.6, paragraph 2:  the reference (Gurung) is incomplete. Corrected

Page 4, section 4, paragraph 1, line 10:  change 'reproduce able ' to 'reproducible'. Done

Page 6, section 7.1, paragraph 3, line 6:  delete repeated words 'acting as'. Done

Page 6, section 7.1, paragraph 3, line 8:  The reference '(Lui)' is incomplete. Corrected

Page 6, section 7.3, paragraph 1, line 7: the reference '(Wu et al.,)' is incomplete. Another Wu et al., 2011 corrected to Wu et al 2021

All of the 15 typos detected by the reviewer have been remedied.  In addition, the reference to Wu et. al. (2011) has been corrected in the bibliography to 2021.  I appreciate the great care taken in reading this report that led to detecting such oversights.

Reviewer 2 Report

Regarding the manuscript entitled "Relationships between Diabetes and the Intestinal Microbial Population"written by Stephen C. Bondy, the following issues could be mentioned:

The topic of the review is interesting and recent data from the literature are discussed. However, most of the cited articles are also literature reviews. 

The author state that "While Type 1 diabetes (T1D) has a more influential genetic source [...], it is increasing equally rapidly as T2D". Please, add a reference.

International Diabetes Federation’s Diabetes Atlas, 2011 could be replaced with the newest version. 

There are a few grammatical errors: "boy weight" instead of bodyweight", "GT1D" and others.

Some of the phyla described have no references attached. 

It is not clear the relevance of Bacillota in relation to diabetes. 

The author describes the influence of one antihyperglycemic drug, while there are pieces of evidence pointing to the role of other classes. This sub-chapter could either be revised by adding more data or removed. eg.   https://doi.org/10.3390/biomedicines10020308

There are references in the text not mentioned in the References section (eg. Tingirkari, 2018). The references should be revised and presented as recommended in the journal author's guidelines. 

Author Response

Regarding the manuscript entitled "Relationships between Diabetes and the Intestinal Microbial Population"written by Stephen C. Bondy, the following issues could be mentioned:

The topic of the review is interesting and recent data from the literature are discussed. However, most of the cited articles are also literature reviews. Emphasis is on most recent works is requested by the journal.  This accounts for inclusion of many recent reviews

The author state that "While Type 1 diabetes (T1D) has a more influential genetic source [...], it is increasing equally rapidly as T2D". Please, add a reference. This phrase is now omitted.

International Diabetes Federation’s Diabetes Atlas, 2011 could be replaced with the newest version. This entry is now amended to 2021.  It has now also been added to the references

There are a few grammatical errors: "boy weight" instead of bodyweight", "GT1D" and others. Type 1.  These errors are corrected and the manuscript re-checked

Some of the phyla described have no references attached. A reference to Matsuoka and Kanai (2015) has been added to the section on Proteobacteria.

It is not clear the relevance of Bacillota in relation to diabetes. The section on  Bacillota is enlarged and now includes a reference to has new ref to(Bojarczuk et al., 2022).

The author describes the influence of one antihyperglycemic drug, while there are pieces of evidence pointing to the role of other classes. This sub-chapter could either be revised by adding more data or removed. eg.   https://doi.org/10.3390/biomedicines1002030830.  This subject has been expanded to include description of a range of other anti-hyperglycemic agents.  I regret that I missed this important 2022 reference.

There are references in the text not mentioned in the References section (eg. Tingirkari, 2018). The references should be revised and presented as recommended in the journal author's guidelines. The reference to Tingirkari (2018) is added and the bibliography checked for other missing references.

Reviewer 3 Report

Author in the review article entitled “Relationships between Diabetes and the Intestinal Microbial Population” evaluated associations between pathogenesis, severity of diabetes and gut microbiota health. It is a well-written and interesting review summarizing this important scientific topic, since scientific interest on gut microbiota-host interactions increase rapidly.

Author at first discussed similarities and differences between T1D and T2D as a short introduction. Afterwards changes in specific Phyla of gut microbiota were discussed with what is especially interesting impact of them on diabetes course and pathophysiology. Lastly, separate paragraph included interventions with established impact on gut microbiota and glucose homeostasis.

Overall, I am satisfied with educational and scientific value of presented article. There are several aspects that I would like to address below.

1)     Author might consider modifying title of the article and include SCFAs since this group of gut microbiota-derived metabolites are mostly discussed in this review.

2)     Authors might consider summarizing changes in specific gut bacteria is association with diabetes in the table to compare them more efficiently and clearly.

3)     It is worth mentioning that there are also other gut microbiota-derived metabolites with many biological effects affecting host’s health apart from SCFAs.

4)     There are some typing and editing errors with word duplication for example in 3rd paragraph of Introduction; 1 paragraph of 2.Type 1 and 2 diabetes, similarities and distinctions”; 1st paragraph of the 3rd section.

Author Response

Overall, I am satisfied with educational and scientific value of presented article. There are several aspects that I would like to address below.

1)     Author might consider modifying title of the article and include SCFAs since this group of gut microbiota-derived metabolites are mostly discussed in this review. To avoid too lengthy title, the focus on SCFAs is now emphasized in the abstract.

2)     Authors might consider summarizing changes in specific gut bacteria is association with diabetes in the table to compare them more efficiently and clearly. Too many other such overview tables have recently been presented. I do not want want to overlap with these too closely.  Instead I constructed a summary figure showing the relationships of the topics discussed.

3)     It is worth mentioning that there are also other gut microbiota-derived metabolites with many biological effects affecting host’s health apart from SCFAs. A new section on this topic has now been added to conclusion.  This includes a new reference (Agus et al., 2021).

4)     There are some typing and editing errors with word duplication for example in 3rd paragraph of Introduction; 1 paragraph of 2.Type 1 and 2 diabetes, similarities and distinctions”; 1st paragraph of the 3rd section. These mistakes have been corrected.  I appreciate the care taken in reading this report that led to detecting such oversights.

Round 2

Reviewer 2 Report

The author adequately responded to all previous comments.

The present review is a valuable manuscript that incorporates the most recent data from the literature.  It is well organized and concisely written.